# Using Whale Optimization Algorithm and Haze Level Information in a Model-Based Image Dehazing Algorithm

**DOI:** 10.3390/s23020815

**Published:** 2023-01-10

**Authors:** Cheng-Hsiung Hsieh, Ze-Yu Chen, Yi-Hung Chang

**Affiliations:** 1Department of Computer Science and Information Engineering, Chaoyang University of Technology, No. 168, Jifong E. Rd., Taichung 413, Taiwan; 2Macronix International Co., No. 19, Lihsin Rd., Science Park, Hsinchu 300, Taiwan

**Keywords:** whale optimization algorithm, model-based image dehazing algorithm, dark channel prior, haze level information, hazy image discriminator, hazy image clustering

## Abstract

Single image dehazing has been a challenge in the field of image restoration and computer vision. Many model-based and non-model-based dehazing methods have been reported. This study focuses on a model-based algorithm. A popular model-based method is dark channel prior (DCP) which has attracted a lot of attention because of its simplicity and effectiveness. In DCP-based methods, the model parameters should be appropriately estimated for better performance. Previously, we found that appropriate scaling factors of model parameters helped dehazing performance and proposed an improved DCP (IDCP) method that uses heuristic scaling factors for the model parameters (atmospheric light and initial transmittance). With the IDCP, this paper presents an approach to find optimal scaling factors using the whale optimization algorithm (WOA) and haze level information. The WOA uses ground truth images as a reference in a fitness function to search the optimal scaling factors in the IDCP. The IDCP with the WOA was termed IDCP/WOA. It was observed that the performance of IDCP/WOA was significantly affected by hazy ground truth images. Thus, according to the haze level information, a hazy image discriminator was developed to exclude hazy ground truth images from the dataset used in the IDCP/WOA. To avoid using ground truth images in the application stage, hazy image clustering was presented to group hazy images and their corresponding optimal scaling factors obtained by the IDCP/WOA. Then, the average scaling factors for each haze level were found. The resulting dehazing algorithm was called optimized IDCP (OIDCP). Three datasets commonly used in the image dehazing field, the RESIDE, O-HAZE, and KeDeMa datasets, were used to justify the proposed OIDCP. Then a comparison was made between the OIDCP and five recent haze removal methods. On the RESIDE dataset, the OIDCP achieved a PSNR of 26.23 dB, which was better than IDCP by 0.81 dB, DCP by 8.03 dB, RRO by 5.28, AOD by 5.6 dB, and GCAN by 1.27 dB. On the O-HAZE dataset, the OIDCP had a PSNR of 19.53 dB, which was better than IDCP by 0.06 dB, DCP by 4.39 dB, RRO by 0.97 dB, AOD by 1.41 dB, and GCAN by 0.34 dB. On the KeDeMa dataset, the OIDCP obtained the best overall performance and gave dehazed images with stable visual quality. This suggests that the results of this study may benefit model-based dehazing algorithms.

## 1. Introduction

Recently, single image haze removal has attracted growing attention in the field of image restoration and computer vision. The haze is mainly due to light scattering from particles in the air. Hazy images generally have reduced contrast and visibility, which degrades the performance of image-based applications. To alleviate this problem, image dehazing or haze removal algorithms are sought. Recently, many dehazing approaches have been reported. These approaches can be roughly divided into two categories: non-model-based and model-based. For non-model-based approaches, thanks to significant advances in deep learning, deep image dehazing models have been used to learn the mapping between hazy images and their ground truth images. This is called end-to-end dehazing. With various structures and learning schemes, many end-to-end deep learning models have been applied to single image haze removal. Some are mentioned below.

An end-to-end framework called DehazeNet was introduced as a pioneering work in this field in [1]. An all-in-one dehazing (AOD) network was proposed in [2]. An adaptive generative adversarial network called CycleGAN was presented in [3]. A deep model called a gated context aggregation network (GCAN) was introduced in [4]. A generic model-agnostic convolutional neural network was reported in [5]. Deep networks for joint transmittance estimation and image dehazing were presented in [6]. A convolutional neural network to remove haze from a single image, involving supervised and unsupervised learning, was presented in [7]. A deep retinex dehazing network with retinex-based decomposition, in which hazy images were decomposed into natural and residual illumination, was proposed in [8]. An end-to-end method of self-guided image dehazing using progressive feature fusion, where the input hazy image was used as the guide image in the dehazing process, was presented in [9]. A deep image dehazing model based on convolutional neural networks was reported in [10]. In the model, ground truth images and high-pass and low-pass filtered images were used in the training phase, which involved fusion and attention mechanisms. This paper concentrates on model-based dehazing methods, thus there are no comments on non-model-based approaches.

Among model-based approaches, the most popular hazy image model is:(1)$$I\left(x\right)=J\left(x\right)t\left(x\right)+A[1-t(x)]$$
where $I\left(x\right)$ is the observed intensity; $J\left(x\right)$ is the scene radiance; A is the global atmospheric light, or simply atmospheric light; $t\left(x\right)=e^{-\beta d(x)}$ is the transmittance, which represents the portion of the light not scattered to the camera; β is the scattering coefficient of the atmosphere; and d(x) is the depth of the scene at position x. The model parameters in Equation (1), A and $t\left(x\right)$, are estimated by various schemes. Among them are learning-based methods. With these methods, researchers try to use deep models to estimate model parameters. For example, in [11], transmittance was found through a multiscale deep neural network; in [12], the atmospheric light map and transmittance were estimated by layer separation; and in [13], a deep learning model called the densely connected pyramid dehazing network was proposed to estimate atmospheric light and transmittance.

The second type of model parameter estimation is based on assumptions or statistical priors. In addition, some reported methods have used optimization algorithms to estimate model parameters. Local uncorrelation was assumed between transmittance and surface shading in [14]; based on this assumption, the model parameters were found. The color line assumption was used in the estimation of model parameters in [15]. A statistical property called dark channel prior (DCP) was observed and applied to estimate the model parameters in [16]. For better dehazing performance, several optimization-based schemes have recently emerged for the estimation of model parameters. An optimization scheme with boundary constraints and contextual regularization was proposed in [17] to estimate transmittance. The hidden Markov random field and the expectation maximization algorithm were presented in [18] to estimate transmittance. A bee colony optimization algorithm was used in [19] to estimate the air light map. The color attenuation prior was introduced and supervised machine learning was used to estimate depth information from the scene in [20]. The optimal transmittance was found by quadratic programming with two scene priors in [21]. A convex optimization was applied to a discrete Harr wavelet transformed model to remove haze from a single image in [22]. A combined radiance–reflectance optimization (RRO) model was proposed in [23] to estimate transmittance. An improved atmospheric scattering model in which a new parameter, called the light absorption coefficient, was introduced to improve the quality of the dehazed images in [24]. LiDAR was used in [25] to generate a grayscale depth image, and then scattering coefficients were found and used to estimate transmittance; finally, a dehazed image was obtained by using the hazy image model. For a recent comprehensive review on image dehazing, see [26].

From the aspect of optimization, none of the above-mentioned studies estimated the optimal scaling factors of the model parameters such as atmospheric light and initial transmittance. In fact, no such estimation has been reported so far. This study attempts to fill the gap by using a metaheuristic optimization algorithm, the whale optimization algorithm (WOA), with the expectation of achieving better dehazing performance. From the aspect of the dataset, no model-based dehazing research based on assumptions and priors has rarely taken advantage of ground truth (GT) images. Currently, such studies generally use GT images in performance evaluation. In this study, we will find optimal scaling factors for the model parameters by the WOA with the help of GT images. In summary, this study provides an alternative way to apply optimization algorithms in image dehazing research to exploit the benefits of GT images.

In single image dehazing research, almost every haze removal approach uses input hazy images as is. Little research has been done to explore image haze information in the field of haze removal. Images were clustered as hazy or non-hazy by a support vector machine in [27], while four types of classification were investigated in [28]. The mean intensity value and entropy were used in [29] to determine images with homogeneous and heterogeneous haze. However, the schemes mentioned above only concentrated on the classification of hazy images without applying them to image dehazing. A hierarchical density-aware dehazing network that extracted haze density information through a density-aware module was proposed in [30]. An affinity propagation clustering algorithm was used in [31] to divide hazy images into different regions of haze density; for each region, atmospheric light and transmittance were estimated and used in an iterative dehazing process. The use of information about the haze level in [30] was implicit, that is, embedded in the network, while hazy images were divided into regions in [31]. Neither of them explicitly used haze level information in a whole image. To date, haze level information has not been used in hazy image discrimination and clustering. This study proposes hazy image discrimination and clustering schemes to expand the research of image haze removal.

The DCP [16] is a popular method because of its simplicity and effectiveness. However, it suffers from the problems of artifacts, halos, color distortion, and high computational cost. The high computational cost can be reduced by using the guided image filter (GIF) [32] to refine the initial transmittance. To enhance the DCP performance, an improved DCP (IDCP) [33] was proposed, in which adaptive scaling factors were introduced for atmospheric light and initial transmittance. In addition, the GIF setting was changed to further improve efficiency. Although the IDCP significantly improved dehazing performance, the adaptive scaling factors were heuristically obtained by the rule of thumb. To improve the dehazing performance of the IDCP, in this paper we attempt to find optimal scaling factors for the IDCP. For that purpose, we introduce a metaheuristic optimization method called the whale optimization algorithm (WOA) [34]. Note that the WOA requires ground truth (GT) images in a fitness function to find an optimal solution. Consequently, a dataset with pairs of hazy and GT images was employed in the study. Moreover, it has been observed that not every GT image is clear, and hazy GT images significantly affect the performance of the WOA. Therefore, we developed a hazy image discriminator (HID) in this study. Because GT images are not available in real-world applications, we present a hazy image clustering (HIC) method to relieve the requirement for GT images in the WOA. The resulting IDCP with optimized scaling factors is called optimized IDCP (OIDCP) in this paper. In Section 4, the proposed OIDCP is justified by three datasets, RESIDE [35], O-HAZE [36], and KeDeMa [37]. The results show that the OIDCP had superior performance over the five comparison methods (IDCP [33], DCP [14], RRO [23], AOD [2], and GCAN [4]) in objective and subjective evaluations. On the RESIDE dataset, the OIDCP achieved a PSNR of 26.23 dB, which was higher than that of IDCP, DCP, RRO, AOD, and GCAN by 0.81, 8.03, 5.28, 5.60, and 1.27 dB, respectively. On the O-HAZE dataset, the OIDCP obtained a PSNR of 19.53 dB, which was better than IDCP by 0.06 dB, DCP by 4.39 dB, RRO by 0.97 dB, AOD by 1.41 dB, and GCAN by 0.34 dB. On the KeDeMa dataset, the OIDCP showed the best objective performance and subjective visual quality of dehazed images. The results indicate that the proposed OIDCP is feasible and promising. It makes at least three main contributions, as follows:The WOA is introduced in a DCP-based image dehazing framework to search optimal scaling factors for the model parameters, i.e., atmospheric light and initial transmittance, with the help of a dataset with pairs of hazy and GT images. This simplifies the optimization process and is essentially different from the reported methods that optimize atmospheric light and/or transmittance. The application of WOA in this study represents an alternative way to use a metaheuristic optimization algorithm in the field of image dehazing. The benefit of the WOA will be verified in Section 4.A hazy image discriminator (HID) is proposed to distinguish hazy images from clear images. The HID was developed based on haze level information extracted from images. In this study, the proposed HID was used to exclude image pairs with hazy GT images. The resulting dataset was then used in the WOA to find optimal scaling factors in the IDCP. The way in which the HID distinguishes hazy images in this study is new in the field of image haze removal. The HID will be validated in Section 4.A hazy image clustering (HIC) scheme is presented based on haze level information. The HIC relieves the requirement for GT images in the proposed OIDCP to make real-world applications possible. Unlike the haze information, which was used implicitly in [30], the proposed HIC uses it explicitly in this study. Furthermore, unlike hazy information, which was used to segment hazy images in [31], the HIC in this paper processes the hazy image as a whole. In addition, the HIC was used to group hazy images and relieve the requirement for GT images in the IDCP/WOA. To date, no method has been reported to divide hazy images into subsets as the HIC does. The HIC will be confirmed in Section 4.

This paper is organized as follows. Section 2 briefly reviews the DCP, IDCP, and WOA. Section 3 introduces the proposed OIDCP approach. In Section 4 the OIDCP is verified, and the IDCP and four recent dehazing methods are compared. Finally, Section 5 concludes this study.

## 2. Review

This section briefly reviews the DCP described in [16], our previous work on the IDCP in [33], and the WOA in [34]. The IDCP and WOA serve as building blocks in the proposed approach described in Section 3.

### 2.1. DCP Dehazing Algorithm

Here, the DCP is reviewed, with the initial transmittance refined by the GIF [32]. Given a hazy image I in the RGB color space, the implementation steps for the DCP are given below.

Step 1.Find the initial dark channel through a block-based minimum filter by
(2)$$I_{\Omega }^{dark}(x)=\underset{y\in \Omega \left(x\right)}{\min}\underset{c}{\min}\left[I_{c}\left(y\right)\right]$$ where $\Omega \left(x\right)$ is an N×N window centered on x and c∈{R,G,B}.Step 2.Estimate the atmospheric light $A=\alpha \times [A_{R}A_{G}A_{B}]$ by $I_{\Omega }^{dark}(x)$, where α=1. Find the 0.1% pixels with the highest values in $I_{\Omega }^{dark}(x)$. Then trace back to the corresponding pixels in image I and find the pixel with the highest intensity as the estimate of A.Step 3.Calculate the normalized dark channel by
(3)$${\overset{-}{I}}_{\Omega }^{dark}(x)=\underset{y\in \Omega \left(x\right)}{\min}\underset{c}{\min}\left[\frac{I_{c}\left(y\right)}{A_{c}}\right]$$Step 4.Obtain the initial transmittance by
(4)$$\tilde{t}\left(x\right)=1-\beta \times {\overset{-}{I}}_{\Omega }^{dark}(x)$$ where β=0.95.Step 5.Refine the initial transmittance $\tilde{t}\left(x\right)$ by the GIF to obtain the final transmittance $t\left(x\right)$, where the guide image is input image I; window size N=20; and smoothing parameter ϵ=0.001.Step 6.Recover the scene radiance by
(5)$${\hat{J}}_{c}\left(x\right)=\frac{I_{c}\left(x\right)-A_{c}}{\max\left[t_{0,}t\left(x\right)\right]}+A_{c}$$ where $t_{0}=0.1$.

It is known that the DCP has problems including artifacts, halos, color distortion, and high computational cost. An improved DCP (IDCP) was proposed in [33] to deal with these problems.

### 2.2. IDCP Dehazing Algorithm

The IDCP in [33] was proposed to deal with the inherent problems of the DCP in [16], that is, artifacts in sky regions, halos around large depth discontinuities, color distortions in dehazed images, and high computational cost. We observed in [33] that the first three problems could be solved by introducing adaptive scaling factors in the estimation of atmospheric light and initial transmittance. Moreover, the computational cost could be reduced by the GIF setting that uses pixel-based dark channel, large window size, and large smoothing factor. For more details, see [33].

Given image I in the RGB color space, whose range is within $\left[0,1\right]$, the implementation steps of the IDCP are described below.

Step 1.Find the pixel-based dark channel as
(6)$$I_{1}^{dark}(x)=\underset{c}{\min}\left[I_{c}\left(x\right)\right]$$ where c∈{R,G,B}.Step 2.Find the maximum in $I_{1}^{dark}(x)$ and its corresponding pixel in I, $p_{max}$. Then estimate the atmospheric light as $A=\left[A_{R}A_{G}A_{B}\right]=\alpha_{a}\times p_{max}$, where $\alpha_{a}=min[{\left(\mu_{1}\right)}^{0.0975},0.975]$ and $\mu_{1}=mean[I_{1}^{dark}(x)]$.Step 3.Find the normalized 15×15 block-based dark channel as
(7)$${\overset{-}{I}}_{15}^{dark}\left(x\right)=\underset{y\in \Omega \left(x\right)}{\min}\underset{c}{\min}\left[\frac{I_{c}\left(y\right)}{A_{c}}\right]$$
where $\Omega \left(x\right)$ is a 15×15 window centered on x.Step 4.Find the initial transmittance as
(8)$$\tilde{t}\left(x\right)=1-\beta_{a}\times {\overset{-}{I}}_{15}^{dark}(x)$$ where $\beta_{a}=min[{\left(\mu_{\tau }\right)}^{0.325},0.95]$ and $\mu_{\tau }=mean[{\overset{-}{I}}_{15}^{dark}(x)\leq \tau ]$ with τ=0.9.Step 5.Obtain the final transmittance $t\left(x\right)$ through refining $\tilde{t}\left(x\right)$ by the GIF where the guide image is $I_{1}^{dark}(x)$ and the window size N=55 and smoothing parameter ϵ=0.1.Step 6.Recover the scene radiance as
(9)$${\hat{J}}_{c}\left(x\right)=\frac{I_{c}\left(x\right)-A_{c}}{\max\left[t_{0,}t\left(x\right)\right]}+A_{c}$$ where $t_{0}=0.1$.

We previously observed [33] that the problems of the DCP result from the fixed scaling factors of A and $\tilde{t}\left(x\right)$, that is, α=1 and β=0.95. Therefore, adaptive scaling factors were introduced into the IDCP for A and $\tilde{t}\left(x\right)$, that is, $\alpha_{a}$ and $\beta_{a}$. The $\alpha_{a}$ introduced into the IDCP eliminated the color distortion and $\beta_{a}$ eliminated the artifacts in the sky region. Figure 1 shows the effect of $\alpha_{a}$ and the scaling factor of A on a dehazed image. As expected, the color distortion was reduced when $\alpha_{a}$ was used. Figure 2 shows the effect of $\beta_{a}$, the scaling factor of $\tilde{t}\left(x\right)$, on a dehazed image. The artifact problem in the sky region was solved when $\beta_{a}$ was used. Figure 3 shows the effect of the GIF setting on a dehazed image. The halo problem at the edge of the front tree was gone when the GIF setting in the IDCP was used. Though the IDCP significantly advanced the performance of the DCP, it can be further improved using optimal scaling factors because it uses heuristic scaling factors that generally do not provide an optimal solution. Thus, the WOA is introduced into the IDCP to further improve its dehazing performance. The details are given in Section 3.

For comparison, the differences in parameters setting in the DCP, IDCP, and IDCP/WOA are summarized in Table 1; IDCP/WOA is discussed in Section 3. The IDCP/WOA is a fundamental part of this study.

### 2.3. Whale Optimization Algorithm

This section provides a brief review of the WOA [34]. The WOA is a metaheuristic optimization algorithm that mimics the social behavior of humpback whales [34]. It is made up of three stages: encircling prey, spiral bubble-net feeding maneuver, and prey search. Related mathematical equations are described below. For details, see [34].

#### 2.3.1. Encircling Prey

Initially, the best current candidate solution is assumed to be the target prey. According to the best selected search agent, other search agents renew their positions toward the target prey. This encircling prey process is modeled as
(10)$$\vec{D}=\left|\vec{C}\cdot {\vec{X}}^{*}\left(t\right)-\vec{X}(t)\right|$$
(11)$$\vec{X}\left(t+1\right)={\vec{X}}^{*}\left(t\right)-\vec{A}\cdot \vec{D}$$
where t is the current iteration; ${\vec{X}}^{*}$ is the current best position vector; $\vec{X}$ is the position vector; || denotes the absolute operator; and · indicates element-to-element multiplication. $\vec{A}$ and $\vec{C}$ are coefficient vectors and calculated, respectively, as
(12)$$\vec{A}=2\vec{a}\cdot \vec{r}-\vec{a}$$
(13)$$\vec{C}=2\cdot \vec{r}$$
where $\vec{a}$ decays linearly from 2 to 0 as the number of iterations increases, in both the exploration and exploitation stages; and $\vec{r}$ denotes a random vector within $\left[0,1\right]$.

#### 2.3.2. Spiral Bubble-Net Feeding Maneuver

In the WOA, two schemes are devised to simulate the bubble-net behavior. One is to shrink the encircled area, which is done by decreasing $\vec{a}$ in Equation (12). The other is to update the position of $\vec{X}$ spirally, which is achieved by the following equation:(14)$$\vec{X}\left(t+1\right)={\vec{D}}^{\prime }\cdot e^{bl}\cdot \cos\left(2\pi l\right)+{\vec{X}}^{*}\left(t\right)$$
where ${\vec{D}}^{\prime }=\left|{\vec{X}}^{*}-\vec{X}\right|$ denotes the distance between ${\vec{X}}^{*}$ and $\vec{X}$; b is a constant to describe the shape of the spiral; and l is a random number in the interval $\left[-1,1\right]$.

Note that the process involves shrinking the encircled area and spirally updating the position simultaneously. The mathematical model with probability is constructed as follows:(15)$$\vec{X}\left(t+1\right)=\left\{\begin{array}{ll} {\vec{X}}^{*}\left(t\right)-\vec{A}\cdot \vec{D} & if\;p<0.5\\ {\vec{D}}^{\prime }\cdot e^{bl}\cdot \cos\left(2\pi l\right)+{\vec{X}}^{*}\left(t\right) & if\;p\geq 0.5 \end{array}\right.$$
where p is a random number within $\left[0,1\right]$.

#### 2.3.3. Search for Prey

In addition to the spiral bubble-net feeding maneuver described above, humpback whales seek prey at random. This is mathematically modeled as follows:(16)$$\vec{D}=\left|\vec{C}\cdot {\vec{X}}_{rand}-\vec{X}\right|$$
(17)$$\vec{X}\left(t+1\right)={\vec{X}}_{rand}-\vec{A}\cdot \vec{D}$$
where ${\vec{X}}_{rand}$ is randomly selected from the population. When $\left|\vec{A}\right|\geq 1$, $\vec{X}$ is updated by ${\vec{X}}_{rand}$ instead of ${\vec{X}}^{*}$. That is, it allows the WOA to avoid being trapped at a local minimum and search for a global solution. The pseudo-code for the WOA is given in Algorithms 1.

In the proposed approach, ${\vec{X}}^{*}$ is the optimal solution for the two scaling factors in the IDCP, and the structural similarity (SSIM) [38] serves as a fitness function. To calculate the SSIM, a reference image (the GT image) is required in addition to a dehazed image. This requirement is eliminated in the proposed OIDCP. In other words, the WOA merely plays an intermediate role in finding optimal scaling factors and is discarded in the dehazing process. The details are described in Section 3.
**Algorithms 1:** Pseudo-code for WOA.Initialize whale population ${\vec{X}}_{i}$ of $N_{w}$ and maximum number of iterations $t_{max}$

Calculate fitness of each search agent ${\vec{X}}_{i}$
Find initial best search agent ${\vec{X}}^{*}$
while ($t<t_{max}$)

      for each search agent ${\vec{X}}_{i}$
            Update $\vec{a}$, $\vec{A}$, $\vec{C}$, l and p

                  if (p<0.5)

                        if ($\left|\vec{A}\right|<1$)

                              Update ${\vec{X}}_{i}$ by Equation (11)

                        else if ($\left|\vec{A}\right|\geq 1$)

                              Randomly select ${\vec{X}}_{rand}$ from population

                              Update ${\vec{X}}_{i}$ by Equation (17)

                        end if

                  else if (p≥0.5)

                        Update ${\vec{X}}_{i}$ by Equation (14)
                  end if

            end for

      Adjust all ${\vec{X}}_{i}$ if they are out of solution range

      Calculate fitness of all ${\vec{X}}_{i}$
      Update ${\vec{X}}^{*}$ if a better ${\vec{X}}_{i}$ is found

      t=t+1
end while

return ${\vec{X}}^{*}$


## 3. Proposed OIDCP

In this section, the proposed approach called optimized IDCP (OIDCP), is introduced. Section 3.1 describes the motivation for the proposed OIDCP, and Section 3.2 provides details on its implementation.

### 3.1. Motivation

Although the IDCP had much better performance than the DCP, its scaling factors $\alpha_{a}$ and $\beta_{a}$ were heuristically determined. That is to say, it generally does not provide an optimal solution. Thus, this paper introduces the WOA to find the optimal scaling factors in the IDCP. The IDCP with the WOA optimized scaling factors is called IDCP/WOA, which plays an intermediate role in the proposed OIDCP. There are five stages involved in the proposed OIDCP. First, as described in Section 2.2, the WOA requires GT images in the fitness function to find the optimal solution. Thus, a dataset S, as the RESIDE dataset, with pairs of hazy and GT images, is used as the input to the WOA. Second, the dataset S is divided into subsets ${\hat{S}}_{k}$ by the proposed hazy image clustering (HIC) scheme according to the haze level information. Third, a hazy image discriminator (HID) is used to screen hazy GT images in ${\hat{S}}_{k}$ because hazy GT images affect the WOA’s ability to find appropriate solutions and degrade the dehazing performance. The reason is that the WOA uses GT images as a reference to find optimal scaling factors. Consequently, image pairs with hazy GT images in ${\hat{S}}_{k}$ are discarded in the IDCP/WOA. The resulting dataset is denoted as $S_{k}$. Fourth, the subset of $S_{k}$ is put into the IDCP/WOA to find the optimal scaling factors ($\alpha_{k,p^{\prime }}^{*}$ and $\beta_{k,p^{\prime }}^{*}$) for the $p^{\prime }$th image pair. Fifth, the requirement for GT images in the IDCP/WOA is relieved because GT images are not available in real-world applications. To this end, the averages of optimal scaling factors $\alpha_{k,p^{\prime }}^{*}$ and $\beta_{k,p^{\prime }}^{*}$ (${\overset{-}{\alpha }}_{k}^{*}$ and ${\overset{-}{\beta }}_{k}^{*}$) are used in the application of the OIDCP. The five stages and their functions are summarized in Table 2.

### 3.2. OIDCP

This section describes how the WOA is incorporated into the IDCP for the IDCP/WOA dehazing approach. In the DCP/WOA, a GT image is required to obtain an optimal solution of scaling factors for atmospheric light and initial transmittance. Although the WOA is able to find an optimal solution, it cannot be applied directly in the dehazing process, because GT images are not available in real-world applications. To this end, the IDCP/WOA is modified to eliminate the requirement for GT images. The resulting scheme, the optimized IDCP (OIDCP), consists of two stages: preparation and application. The overall block diagram of the proposed OIDCP is given in Figure 4.

Given a dataset that has pairs of hazy and GT images, such as the RESIDE dataset in [35], there are four steps in the OIDCP preparation stage. Denote the given dataset as $S=\left\{\left(I_{i}^{h},I_{j}^{g}\right),for\;1\leq i\leq N_{h},1\leq j\leq N_{g}\right\}$, where $I_{i}^{h}$ is a hazy image and $I_{j}^{g}$ is its associated GT image; $N_{h}$ and $N_{g}$ are the number of hazy and GT images, respectively. Note that $N_{h}$ and $N_{g}$ are different, which implies that one GT image may relate to many hazy images. This is true in the RESIDE dataset and other datasets. With S, the preparation steps are described below.
Step 1.The HIC (described in Section 3.2.1) is performed to divide $I_{i}^{h}$ into K subsets. The subsets are denoted as ${\hat{S}}_{k}$ for 1≤k≤K, where ${\hat{S}}_{k}=\left\{\left({\tilde{I}}_{k,l}^{h},{\tilde{I}}_{k,m}^{g}\right),for\;1\leq l\leq {\tilde{N}}_{k,h}\;and\;1\leq m\leq {\tilde{N}}_{k,g}\right\}$ and ${\tilde{N}}_{k,h}$ and ${\tilde{N}}_{k,g}$ are the number of hazy and GT images, respectively in haze level (HL) k.Step 2.For each ${\hat{S}}_{k}$, the HID (described in Section 3.2.2) is performed to select clear GT images $I_{k,n}^{g}$ from ${\tilde{I}}_{k,m}^{g}$. The obtained $I_{k,n}^{g}$ with corresponding hazy images $I_{k,p}^{h}$ form a set $S_{k}$ with image pairs $\left(I_{k,p}^{h},I_{k,n}^{g}\right)$.Step 3.Given a percentage p%, $N_{s}$ hazy images are randomly chosen from $I_{k,p}^{h}$, for 1≤k≤K. With their associated GT images, the selected image set $S_{k}^{s}=\left\{\left(I_{k,p^{\prime }}^{h,s},I_{k,n^{\prime }}^{g,s}\right),for\;1\leq p^{\prime }\leq N_{k,h},1\leq n^{\prime }\leq N_{k,g},1\leq k\leq K\right\}$ is obtained, where $N_{k,h}$ and $N_{k,g}$ are the number of hazy and GT images, respectively, and $N_{s}=\sum_{k=1}^{K}N_{k,h}$.Step 4.Set $S_{k}^{s}$ is used in the IDCP/WOA to find the optimal scaling factors $\alpha_{k,p^{\prime }}^{*}$ and $\beta_{k,p^{\prime }}^{*}$. Then the averages of $\alpha_{k,p^{\prime }}^{*}$ and $\beta_{k,p^{\prime }}^{*}$ (${\overset{-}{\alpha }}_{k}^{*}$ and ${\overset{-}{\beta }}_{k}^{*}$) for HL k are found and used in the OIDCP application stage.


There are two steps in the OIDCP application stage. First, an image is put into the HIC to determine its HL, say k. Second, the OIDCP is performed to obtain the dehazed image. It should be mentioned that using ${\overset{-}{\alpha }}_{k}^{*}$ and ${\overset{-}{\beta }}_{k}^{*}$ does not require GT images in the application stage. Because the application stage is simple and clear, no further explanation is given here.

#### 3.2.1. Hazy Image Clustering

Until now, haze level information in images has rarely been explicitly utilized in the field of image haze removal. In this study, a hazy image clustering (HIC) scheme is presented to fill this research gap. The HIC is motivated by the properties of the dark channel prior described in [16]. In that study, He et al. observed that in a patch the pixel value of at least one of the RGB components in a haze-free image, without white objects or sky regions, was zero or very low. For a hazy image, the dark channel is not close to zero and is brighter. This statistical property is called the dark channel prior [16]. The dark channel can be obtained by a minimum filter. Figure 5 provides an example showing the property of the dark channel prior, using the 15 × 15 minimum filter. Figure 5a shows a haze-free image, whose dark channel is shown in Figure 5b. The corresponding hazy image is shown in Figure 5c, whose dark channel is shown in Figure 5d. As seen in Figure 5b, the dark channel for the haze-free image in Figure 5a is very dark except for the sky regions. However, the hazy image in Figure 5c has a brighter dark channel than that in Figure 5b, as described above.

The results in Figure 5 suggests that the average of the dark channel can be used as a measure of haze level (HL) if white objects and sky regions are excluded to avoid bias. This can be achieved by introducing a threshold. The proposed HIC involves three steps. First, a threshold τ is used to exclude white objects and sky regions in the dark channel calculation of a hazy image. Second, the truncated average value of the dark channel in step 1 is calculated and used as the HL measure. Third, the hazy image is assigned to a cluster (subset) according to its HL. Given a hazy image I, the proposed HIC is implemented as follows:Step 1.Find the 15×15 block-based dark channel as
(18)$$I_{15}^{dark}(x)=\underset{y\in \Omega \left(x\right)}{\min}\underset{c}{\min}\left[I_{c}\left(y\right)\right]$$
where c∈{R,G,B} and $\Omega \left(x\right)$ is a 15×15 window centered on x.Step 2.Calculate the truncated average of $I_{15}^{dark}(x)$ as
(19)$${\tilde{\mu }}_{dc}\left(\tau \right)=mean[I_{15}^{dark}(x)\leq \tau ]$$
where 0<τ≤1 is a user-defined threshold.Step 3.Assign I to a predefined HL k, for 1≤k≤K, according to ${\tilde{\mu }}_{dc}\left(\tau \right)$, where K is the number of HLs.


In the proposed OIDCP, the HIC is used to divide hazy images into subsets in the preparation stage and to assign the HL index to the input image in the application stage.

#### 3.2.2. Hazy Image Discriminator

The RESIDE dataset [35] was used in the experiment in this study. Note that in the RESIDE dataset, 35 synthesized hazy images are generated by one GT image with different model parameters in Equation (1). Furthermore, we observed that (i) not every GT image was clear and (ii) clear GT images improved the dehazing performance of the IDCP/WOA. This implies that selecting clear GT images plays a decisive role in the performance of the IDCP/WOA, because the WOA attempts to seek $\alpha_{k,i}^{*}$ and $\beta_{k,i}^{*}$ using a GT image as a reference. In other words, the dehazed image by the IDCP/WOA is hazy if the GT image is hazy, and vice versa. To verify this, two examples are given in Table 3, which show the IDCP/WOA results with clear and hazy GT images; the IDCP result is also given for comparison. The results indicate that the dehazed image by the IDCP/WOA is close to the GT image. When the GT image is clear, the IDCP/WOA dehazed image is also clear, as shown in the first row of Table 3. When the GT image is hazy, the dehazed image is accordingly hazy, as in the second row of Table 3. The result explains that selecting clear GT images is crucial to the IDCP/WOA.

As described above, finding clear GT images was fundamental to the success of this study. Therefore, a haze image discriminator (HID) was developed. The motivation for the HID is based on two observations. First, a GT image generally has less haze, thus ${\tilde{\mu }}_{dc}\left(0.4\right)$ can serve as the first check for ${\tilde{I}}_{k,m}^{g}$. When the first check fails, a further check is performed. Second, ${\tilde{\mu }}_{dc}\left(\tau \right)$ changes little with a clear ${\tilde{I}}_{k,i}^{g}$ as τ increases, whereas a hazy ${\tilde{I}}_{k,i}^{g}$ does not. This observation lays the basis for the second check of ${\tilde{I}}_{k,i}^{g}$. Given image I, the implementation steps of the HID are described below.
Step 1.Find the 15×15 block dark channel by
(20)$$I_{15}^{dark}(x)=\underset{y\in \Omega \left(x\right)}{\min}\underset{c}{\min}\left[I_{c}\left(y\right)\right]$$
where c∈{R,G,B} and $\Omega \left(x\right)$ is a 15×15 window centered on x.Step 2.Calculate the truncated average of $I_{15}^{dark}(x)$ by
(21)$${\tilde{\mu }}_{dc}\left(\tau \right)=mean[I_{15}^{dark}(x)\leq \tau ]$$
where 0<τ≤1 is a user-defined threshold.Step 3.Check if the inequality ${\tilde{\mu }}_{dc}\left(\tau \right)<\eta$ holds, where η is a user-defined threshold. If ${\tilde{\mu }}_{dc}\left(\tau \right)<\eta$ is true, then image I is considered as clear. Otherwise, go to step 4.Step 4.Calculate the difference ${∆\tilde{\mu }}_{dc}={\tilde{\mu }}_{dc}\left(\tau_{1}\right)-{\tilde{\mu }}_{dc}\left(\tau \right)$ and check if the inequality ${∆\tilde{\mu }}_{dc}<\epsilon$ holds, where $\tau_{1}>\tau$ and ϵ is a user-defined threshold. If ${∆\tilde{\mu }}_{dc}<\epsilon$ is true, then image I is considered as clear; otherwise, it is hazy.


The three images shown in Table 4 are used to demonstrate how the proposed HID works, where τ=0.4, η=0.1 and $\tau_{1}=0.9$. Image $I_{1}$ is considered a clear GT image, since ${\tilde{\mu }}_{dc}\left(0.4\right)=0.0756<0.1$. For image $I_{2}$, since ${\tilde{\mu }}_{dc}\left(0.4\right)=0.2461>0.1$, $I_{2}$ needs a further test using $∆{\tilde{\mu }}_{dc}$. Because $∆{\tilde{\mu }}_{dc}={\tilde{\mu }}_{dc}\left(0.9\right)-{\tilde{\mu }}_{dc}\left(0.4\right)=0.0281<0.1$, $I_{2}$ is considered to be a clear GT image. Image $I_{3}$ is determined to be a hazy GT image because ${\tilde{\mu }}_{dc}\left(0.4\right)=0.1917>0.1$ and $∆{\tilde{\mu }}_{dc}=0.1454>0.1$. Table 4 indicates that the HID result is consistent with the visual result.

#### 3.2.3. Averages of ${\overset{-}{\alpha }}_{k}^{*}$ and ${\overset{-}{\beta }}_{k}^{*}$

With the selected image dataset $S_{k}^{s}=\left\{\left(I_{k,p^{\prime }}^{h,s},I_{k,n^{\prime }}^{g,s}\right),for\;1\leq p^{\prime }\leq N_{k,h},1\leq n^{\prime }\leq N_{k,g},1\leq k\leq K\right\}$, the WOA searched the optimal scaling factors $\alpha_{k,p^{\prime }}^{*}$ and $\beta_{k,p^{\prime }}^{*}$ in the IDCP, where $I_{k,p^{\prime }}^{h,s}$ was the input image and $I_{k,n^{\prime }}^{g,s}$ was the reference GT image. The parameters $N_{w}=10$ and $t_{max}=10$ are set in the WOA. The fitness function used here is structural similarity (SSIM) [38]. Given $I_{k,p^{\prime }}^{h,s}$, for each iteration t, the scaling factors $\alpha_{k,p^{\prime }}$ and $\beta_{k,p^{\prime }}$ were found by the WOA. Then the two scaling factors were applied to the IDCP to find a dehazed image ${\hat{I}}_{k,p^{\prime }}^{d}$. The SSIM was calculated between ${\hat{I}}_{k,p^{\prime }}^{d}$ and $I_{k,n^{\prime }}^{g,s}$, which was then used to update $\alpha_{k,p^{\prime }}$ and $\beta_{k,p^{\prime }}$ obtained by the WOA. This process continues until $t_{max}$ is reached. The resulting $\alpha_{k,p^{\prime }}$ and $\beta_{k,p^{\prime }}$ were the optimal scaling factors $\alpha_{k,p^{\prime }}^{*}$ and $\beta_{k,p^{\prime }}^{*}$ for $I_{k,p^{\prime }}^{h,s}$. Figure 6 shows the block diagram of the search of $\alpha_{k,i}^{*}$ and $\beta_{k,i}^{*}$ by the IDCP/WOA. For each HL k, the averages of $\alpha_{k,p^{\prime }}^{*}$ and $\beta_{k,p^{\prime }}^{*}$ are respectively calculated as
(22)$${\overset{-}{\alpha }}_{k}^{*}=\frac{1}{N_{k}}\sum_{p^{\prime }=1}^{N_{k}}\alpha_{k,p^{\prime }}^{*}$$
and
(23)$${\overset{-}{\beta }}_{k}^{*}=\frac{1}{N_{k}}\sum_{p^{\prime }=1}^{N_{k}}\beta_{k,p^{\prime }}^{*}$$

Scaling factors ${\overset{-}{\alpha }}_{k}^{*}$ and ${\overset{-}{\beta }}_{k}^{*}$ for HL k are employed in the application stage of the OIDCP.

## 4. Results and Discussion

The preparation of training data for the experiment is described in Section 4.1, where the parameter settings of the HID and the HIC are discussed. In Section 4.2, the IDCP with three metaheuristic optimization algorithms (WOA [33], BRO [39], and MRFO [40]) are investigated to explain why the WOA is selected in this study. In Section 4.3, the effect of the number of image pairs ($N_{s}$) in training set and the number of HLs (K) on the IDCP/WOA is studied, and $N_{s}$ and K are determined. In Section 4.4, the OIDCP and IDCP are compared to show the superiority of the optimized scaling factors (${\overset{-}{\alpha }}_{k}^{*}$ and ${\overset{-}{\beta }}_{k}^{*}$) over the heuristic scaling factors ($\alpha_{a}$ and $\beta_{a}$). In Section 4.5, the proposed OIDCP is justified by three datasets (RESIDE [35], O-HAZE [36], and KeDeMa [37]) and compared with the IDCP and four recent dehazing algorithms.

### 4.1. Training Data Preparation

The RESIDE dataset is a large-scale benchmark for single image dehazing algorithms that includes 8970 GT images and 313,950 synthetic hazy images. As described in Section 3.2.2, the IDCP/WOA performance is significantly affected by GT images; that is, better results are obtained when clear GT images are used. Therefore, preparing an appropriate training set is necessary to ensure better performance by the IDCP/WOA. Preparing the training data involves two steps: image pair selection by the HID and determination of the number of HLs (K) and hazy images in HL *k* ($N_{k}$) in the HIC.

#### 4.1.1. Image Pair Selection by the HID

In the proposed HID, clear GT images are selected, together with their corresponding synthetic hazy images, and a selected dataset is formed. In the HID, the user-defined thresholds τ, η, and $\tau_{1}$ should be set. As a general rule, τ=0.4 and $\tau_{1}=0.9$ were used in the experiments. Note that threshold η significantly affects the result, thus, its effect was investigated. The numbers of GT images ($N_{g}$) and hazy images ($N_{h}$) with various η values are shown in Table 5, which also shows the ratio $N_{g}/8970$ as a percentage (R%). In Table 5, a small gap of $N_{h}$ falls between η=0.05 and η=0.025; consequently, the dataset with η=0.05 was selected, denoted as S(0.05), which was used as a training set in the following experiments. Table 5 indicates that the proposed HID considered 5633 GT images hazy, thus they were excluded from the RESIDE dataset. That is, only 36.09% of the GT images were kept and used in the experiments.

#### 4.1.2. Determination of K and $N_{k}$

Note that the selected dataset by the HID had dim images with generally low pixel values, thus confusing the calculation of ${\tilde{\mu }}_{dc}\left(\tau \right)$. Because these images degrade the performance of the IDCP/WOA, they were excluded from dataset S(0.05). Furthermore, it was observed that the haze level measure ${\tilde{\mu }}_{dc}\left(0.9\right)$ in S(0.05) had a Gaussian distribution, as shown in Figure 7. Therefore, the training dataset in the experiment was randomly selected according to the distribution of ${\tilde{\mu }}_{dc}\left(0.9\right)$.

Note that there are few hazy images of S(0.05) with the lowest HL (k=1) and the highest HL (k=K), as shown in Figure 7. To ensure enough samples, we fixed the lowest HL at ${\tilde{\mu }}_{dc}\left(0.9\right)\leq 0.15$ and the highest HL at ${\tilde{\mu }}_{dc}\left(0.9\right)>0.6$. These boundaries are marked in Figure 7. The rest of the intervals were equally divided as (0.6-0.15)/(K-2), where K is the number of HLs. The HLs for different K are listed in Table 6, where the percentage p% of each HL is also given. Table 6 was used in the following experiment, in which the hazy images were randomly selected according to p%.

### 4.2. IDCP with WOA, MMFOA, BRO, and MRFO

The purpose of the experiment described in this section was to investigate the effects of different metaheuristic optimization algorithms on the dehazing performance of the IDCP. In other words, a comparison was made between the optimal scaling factors obtained from different metaheuristic optimization algorithms. In the experiment, four metaheuristic optimization algorithms were considered: WOA [33], MMFOA [39], BRO [40], and MRFO [41]. In the WOA, the parameters were set as $N_{w}=10$ and $t_{max}=10$. In the MMFOA, the parameters were set as area_limit=10, life_time=15, transfer_rate=10, and iterations=2. In the BRO, the parameters were set as N=50, maxiter=3, and MaxFault=5. In the MRFO, the parameters were set as MaxIteration=100, PopSize=10, and FunIndex=2. The terms used in these parameter settings were the same as the source codes provided in the related research. In the experiment, 4000 image pairs were randomly selected from S(0.05). PSNR and SSIM were used to evaluate the performance. Table 7 shows the comparison results, indicating that the IDCP/WOA had the best performance, which was superior to the IDCP/MMFOA, IDCP/BRO, and IDCP/MRFO by 0.256, 0.529, and 0.317 dB, respectively. Thus, the IDCP/WOA was selected in this study.

### 4.3. Effect of $N_{s}$ and K on IDCP/WOA

The effect of $N_{s}$ and K on the IDCP/WOA with dataset S(0.05) was investigated. In the experiment, $N_{s}$ image pairs were randomly selected from S(0.05) and were then clustered into K subsets by the proposed HIC. For each cluster, the optimal scaling factors were found by the IDCP/WOA, using $N_{w}=10$ and $t_{max}=10$ in the WOA. Table 8 shows the average PSNR for each combination with various values of $N_{s}$ (2000, 3000, 4000, and 5000) and K (5, 6, 7, 8, 9, and 10).

Note that HL 1 (k=1) and HL K (k=K) were fixed, as previously described. In addition, the number of hazy images was different for each K. For a fair comparison, the PSNR of k=2 in each K was investigated because some part of the identical hazy images was retained when K was changed. For example, in the case of k=2, K=6, part of the hazy images was from those with k=2, K=5, because the cases k=2, K=5 and k=2, K=6 had the same boundary, ${\tilde{\mu }}_{dc}\left(0.9\right)=0.15$, as shown in Table 3. In other words, the hazy images in the case of k=2, K=5 were divided into two parts, one part for k=2, K=6 and the other part for k=3, K=6. As K increases, fewer hazy images are kept in k=2. This suggests that optimal scaling factors generally have less variance in k=2; consequently, better performance can be expected. This idea is verified in the results shown in Table 4. For example, when $N_{s}=4000$, the PSNR for k=2 with values of K from 5 to 10 was 30.39, 30.48, 30.50, 30.65, 30.71, and 30.78 dB, respectively. This result suggests that PSNR generally increases as K increases. This trend is also generally true for other $N_{s}$. Furthermore, the results in Table 8 indicate that PSNR decreases as HL k increases. In other words, the dehazing performance of the IDCP/WOA degrades as k increases. In the following sections, the combination of K=10 and $N_{s}=4000$ is used because it has a slightly better result, with an average PSNR of 28.20 dB. The case of K=10 and $N_{s}=4000$ was also used to find the scaling factors ${\overset{-}{\alpha }}_{k}^{*}$ and ${\overset{-}{\beta }}_{k}^{*}$, as described in Section 3.2.3.

### 4.4. Comparison of IDCP and Proposed OIDCP

In this section, the proposed OIDCP is compared with the IDCP. The experiments were conducted to demonstrate that the optimized scaling factors by the WOA in the OIDCP are better than the heuristic scaling factors in the IDCP. In the experiment, 10,000 hazy images were randomly selected from S(0.05), excluding the training data used in the IDCP/WOA. Note that each objective assessment has a preference for a different aspect. Therefore, seven objective assessments were considered for a fair comparison: SSIM, PSNR, BRISQUE [42], ILNIQE [43], TMQI [44], FSITM [45], and F&T, the average of TMQI and FSITM. Among the seven performance indices, five are full reference assessments that require GT images in the evaluation (SSIM, PSNR, TMQI, FSITM, and F&T), while two are no-reference assessments that do not need GT images (BRISQUE and ILNIQE). The overall performance index is the average rank ($\overset{-}{R}$) of the seven evaluations. The objective results are given in Table 9, where the arrow indicates the direction of better performance, and the corresponding rank is shown in parentheses. Table 9 shows that the IDCP and OIDCP have the same SSIM, and the IDCP has better results on BRISQUE and ILNIQE, and worse results on the rest of the performance indices. The results indicate that the overall performance index $\overset{-}{R}$ is highest for the proposed OIDCP. This implies that the OIDCP with optimized scaling factors ${\overset{-}{\alpha }}_{k}^{*}$ and ${\overset{-}{\beta }}_{k}^{*}$ has better performance than the IDCP with heuristic scaling factors $\alpha_{a}$ and $\beta_{a}$, as expected. It should be mentioned that the average PSNR for the OIDCP was 26.23 dB, which is 1.97 dB less than that of the IDCP/WOA (28.20 dB). The PSNR loss is the price paid to eliminate the requirement for GT images in the IDCP/WOA. This compromise should be considered in further research to improve the OIDCP.

For subjective comparison, 10 images were selected from different HLs to compare the visual quality of dehazed images from the IDCP and OIDCP. The dehazed images are shown in Table 10, where the GT image ($I_{g}$) its corresponding hazy image ($I_{h}$), and the corresponding PSNR are given. Notation $I_{k}$ in Table 10 stands for the hazy image whose HL is k, e.g., $I_{1}$ is an image in HL 1 (k=1). In Table 10, the haze level from $I_{1}$ to $I_{10}$ increases as k increases, which is consistent with the visual result. This suggests that the proposed HIC is appropriate for clustering hazy images. Moreover, there are no hazy GT images in Table 10. This indicates that the proposed HID works well in the selection of clear GT images. As shown in Table 10, the IDCP obtained satisfactory dehazed results without halos, artifacts, or color distortion. However, the proposed OIDCP shows better visual quality of the dehazed image than the IDCP. The superiority of the OIDCP over the IDCP is confirmed.

### 4.5. Comparison of OIDCP, IDCP, DCP, RRO, AOD, and GCAN

In this section, the proposed OIDCP is verified and compared with the IDCP and four recent dehazing methods, DCP [16], RRO [23], AOD [2], and GCAN [4]. The comparison methods are objectively and subjectively evaluated in the following.

#### 4.5.1. Results for RESIDE Dataset

In this section, the proposed OIDCP and comparison methods are objectively evaluated using the RESIDE dataset. In the experiment, 10,000 hazy images were randomly selected from S(0.05), excluding the training data used in the IDCP/WOA. For the OIDCP, the application stage in Figure 4 was executed in the experiment. As in Section 4.4, the seven objective assessments were used in the objective evaluation and $\overset{-}{R}$ was used as the overall performance index. Table 11 shows the results for the proposed OIDCP and comparison methods. The table indicates that based on $\overset{-}{R}$, the methods from best to worst are OIDCP, IDCP, GCAN, RRO, AOD, and DCP. The OIDCP achieved the best PSNR result of 26.23 dB, which is better than IDCP by 0.81 dB, DCP by 8.03 dB, RRO by 5.28, AOD by 5.6 dB, and GCAN by 1.27 dB. The result shows that the OIDCP is better than the IDCP, which is better than the DCP. These three methods are all based on the dark channel prior originated in [16]. The result validates that heuristic scaling factors (IDCP) are better than fixed ones (DCP), but both are worse than optimized ones (OIDCP). In addition, the result indicates that optimizing the scaling factors in the OIDCP is better than optimizing the combined radiance–reflection in the RRO. The OIDCP is also objectively superior to end-to-end deep image dehazing models (AOD and GCAN). In summary, our OIDCP works well on the RESIDE dataset.

Subjective comparisons of the OIDCP, IDCP, DCP, RRO, AOD, and GCAN are provided here. Ten images in different HLs were used to compare the subjective quality of dehazed images by each method. In Table 12, the subjective results show that the IDCP has a satisfactory result and the DCP suffers from the problems of artifacts ($I_{2}$, $I_{6}$), halos ($I_{2}$, $I_{3}$, $I_{6}$), and color distortion ($I_{2}$, $I_{3}$, $I_{6}$, $I_{7}$, $I_{9}$). The RRO has the problem of color distortion ($I_{2}$, $I_{3}$, $I_{7}$, $I_{9}$), and the AOD also tends to have a color distortion problem ($I_{1}–I_{3}$) and is unable to remove haze ($I_{4}$–$I_{10}$). The GCAN has a color oversaturation problem ($I_{1}$, $I_{5}$, $I_{8}$). On the contrary, our OIDCP appropriately removes haze from all images when comparing $I_{g}$. Again, the result in Table 12 implies that the proposed OIDCP is superior to the comparison methods. The result is consistent with the objective evaluation in Table 11.

#### 4.5.2. Results for O-HAZE Dataset

In this section, the proposed OIDCP is further verified by the O-HAZE dataset [36]. The O-HAZE dataset consists of 45 clear images and their corresponding hazy images. The hazy images were generated by professional haze machines, rather than being artificially generated, as in [35]. The objective and subjective results are investigated below.

The objective result for the OIDCP and comparison methods is given in Table 13. It shows that the RRO has the best $\overset{-}{R}$, followed by OIDCP, IDCP, GCAN, AOD, and DCP follows. Although the RRO has the best $\overset{-}{R}$, it does not have good SSIM and PSNR, while the other evaluation indices have a preference for the RRO. On the contrary, the proposed OIDCP shows the best performance on the SSIM and PSNR, whereas it has a worse result than the RRO on other indices. This implies that better visual quality can be expected with the OIDCP because the results from full reference assessments, e.g., PSNR, are more reliable than no-reference assessments, e.g., BRISQUE, ILNIQE. Moreover, the IDCP generally has a better result than the DCP. Again, heuristic scaling factors in the IDCP have a better result than fixed scaling factors in the DCP, and the optimized scaling factors in the OIDCP outperform those in the IDCP. The AOD generally has low values on performance indices, while the GCAN has below average performance. For the two end-to-end image dehazing models, poor performance might result from learning inappropriate mapping from image pairs with hazy GT images.

The result of the subjective evaluation is shown in Table 14 for the OIDCP and the comparison methods. As shown in the table, five images were selected from the O-HAZE dataset to show the visual quality of dehazed images using the comparison methods. The dehazed images in all methods are not good when compared with $I_{g}$. In other words, all six dehazing methods had difficulty appropriately removing the haze. The DCP generally produced dim dehazed images with color distortion, whereas the IDCP was better. The RRO had better contrast in images 2 to 4 and poor results in images 1 and 5. The AOD had less dehazed images with color distortion, whereas the GCAN has better results than the AOD. Among all methods, the OIDCP constantly provided better visual quality of dehazed images, as mentioned above. In summary, all six methods produced dim dehazed images and color distortion to some degree in the given examples. The result suggests that both the proposed OIDCP and the comparison methods are not able to handle images, that is, the O-HAZE dataset, whose haze was generated from a haze machine.

#### 4.5.3. Results with KeDeMa Dataset

In this section, the KeDeMa [37] dataset is used to further evaluate the proposed OIDCP and the comparison methods. The KeDeMa dataset includes 25 natural hazy images with different scenarios, and no GT images. Thus, only two no-reference performance indices, BRISQUE and ILNIQE, were applied in the experiment. The objective and subjective results are given below.

The objective results for the comparison methods are given in Table 15, which shows that the proposed OIDCP and the DCP had the best $\overset{-}{R}$, followed by RRO, IDCP, AOD, and GCAN. The OIDCP and GCAN had comparatively stable ranking in BRISQUE and ILNIQE, whereas the other comparison methods had a large jump in the ranking. This implies that the OIDCP can continually provide better visual quality of dehazed images than the other methods. This is verified in the following.

The subjective result is shown in Table 16. As shown in the table, five images were selected from the KeDeMa dataset for visual quality comparison. It indicates that the DCP has problems with halos (images 2 to 5) and color distortion (image 1), whereas the IDCP reduces these problems. The AOD generally has less dehazed images, except image 1, and the RRO generally has color distortion in dehazed images, whereas the GCAN has color distortion (image 4) and artifacts (lower right corner in image 5). In contrast, the proposed OIDCP constantly gives better results than the other methods, as expected. This indicates that the objective results are not consistent with the subjective results. For example, the DCP has the poorest dehazed images, but the best $\overset{-}{R}$. In other words, the performance indices, BRISQUE and ILNIQE, are not appropriate to measure the visual quality of dehazed images for the KeDeMa dataset.

## 5. Conclusions

This paper presents an optimized IDCP (OIDCP) dehazing algorithm using the WOA and haze level information. In the proposed OIDCP, the WOA was used to find the optimal scaling factors for atmospheric light and initial transmittance in the IDCP. The IDCP with the WOA scheme was termed IDCP/WOA. Based on the haze level information, the hazy image discriminator (HID) and hazy image clustering (HIC) were developed. The HID filters out hazy GT images, whereas the HIC divides hazy images into clusters according to the haze level information. In this study, the IDCP/WOA played an intermediate role, and was used to find the optimal scaling factors for atmospheric light and initial transmittance with the help of GT images. To eliminate the requirement for GT images in the WOA, the averages of the optimal scaling factors for each haze level were calculated and used in real-world applications. The final dehazing algorithm was called the optimized IDCP (OIDCP), that is, the IDCP using the averages of optimal scaling factors by the IDCP/WOA. The proposed OIDCP was verified and compared with the IDCP and four recent dehazing methods, DCP, RRO, AOD, and GCAN. Three datasets were used to justify the proposed OIDCP in the experiment, the RESIDE, O-HAZE, and KeDeMa datasets. With the RESIDE dataset, the OIDCP reached a PSNR of 26.23 dB, which was superior to IDCP (0.81 dB), DCP (8.03 dB), RRO (5.28 dB), AOD (5.60 dB), and GCAN (1.27 dB). With the O-HAZE dataset, the OIDCP had a PSNR of 19.53 dB, which was better than IDCP (0.06 dB), DCP (4.39 dB), RRO (0.97 dB), AOD (1.41 dB), and GCAN (0.34 dB). With the KeDeMa dataset, the OIDCP obtained the best overall performance and provided dehazed images with stable visual quality. The results suggest that the proposed approach could be applied to model-based dehazing algorithms to benefit the dehazing performance. In addition, this study provides an alternative way to use metaheuristic optimization algorithms in model-based haze removal methods, which is to search for optimal scaling factors for model parameters instead of the model parameters itself. There are at least two ways to further improve the proposed OIDCP. First, the dehazing performance needs to be improved on hazy images, such as in the O-HAZE dataset, because the visual quality of dehazed images is not satisfactory. Second, the way to find scaling factors for atmospheric light and initial transmittance can be improved, because the loss of PSNR between the IDCP/WOA (28.20 dB; $K=10\;and\;N_{s}=4000$) and the OIDCP (26.23 dB) was 1.97 dB. Further research on this topic will emphasize these two aspects.

## Figures and Tables

**Figure 1 sensors-23-00815-f001:**
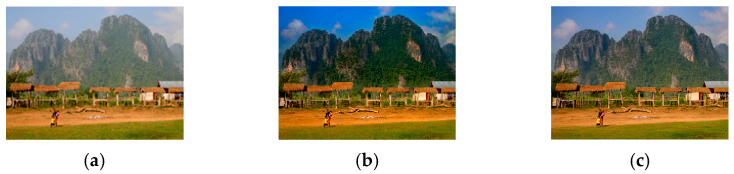
Example showing effects of scaling factor of A on a dehazed image: (**a**) hazy image; (**b**) DCP (α=1); (**c**) IDCP ($\alpha_{a}=0.906$).

**Figure 2 sensors-23-00815-f002:**
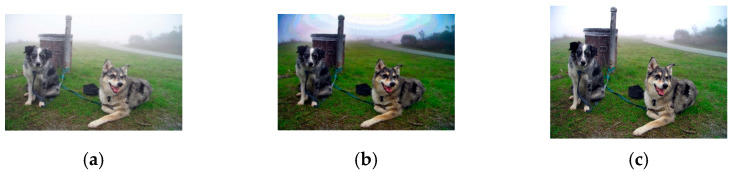
Example showing effects of scaling factor of $\tilde{t}\left(x\right)$ on a dehazed image: (**a**) hazy image; (**b**) DCP (β=0.95); (**c**) IDCP ($\beta_{a}=0.658$).

**Figure 3 sensors-23-00815-f003:**
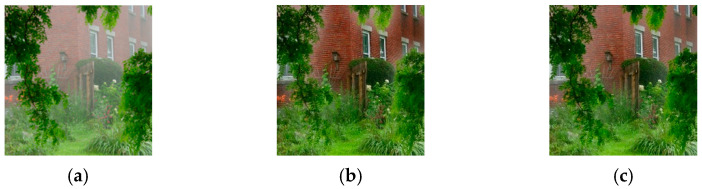
Example showing effects of GIF setting on a dehazed image: (**a**) hazy image; (**b**) DCP; (**c**) IDCP.

**Figure 4 sensors-23-00815-f004:**
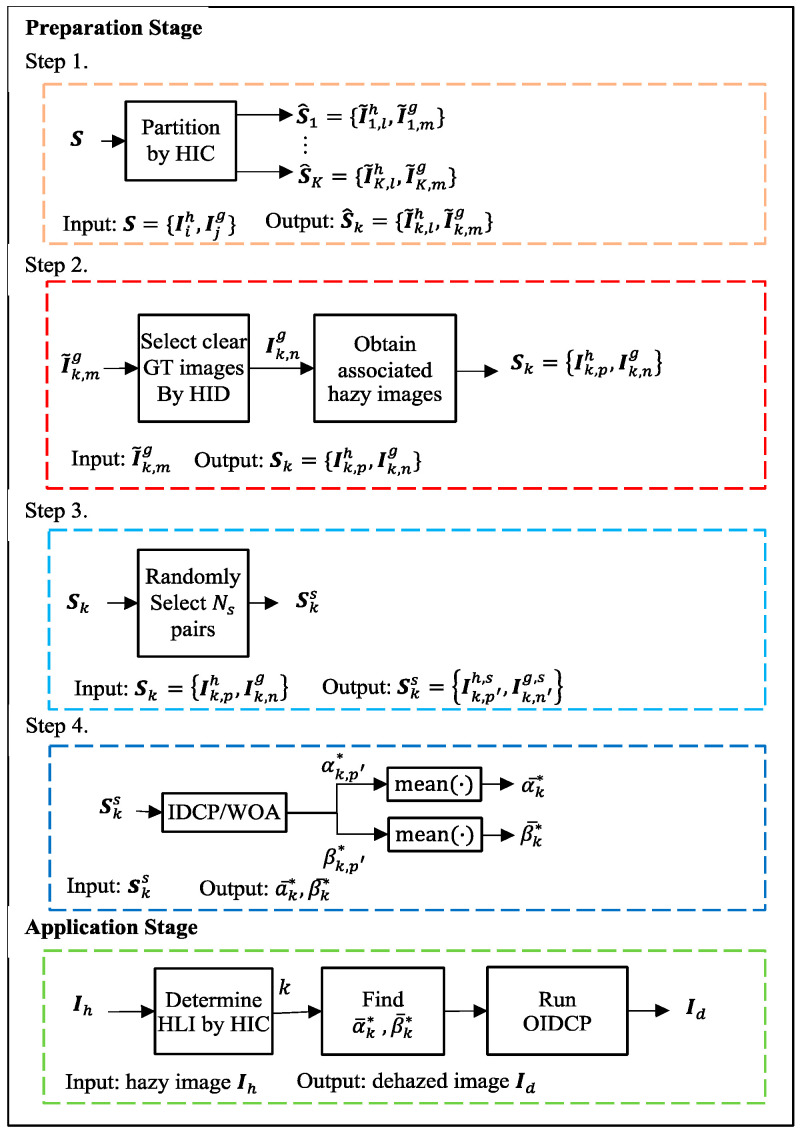
Overall block diagram of proposed OIDCP.

**Figure 5 sensors-23-00815-f005:**
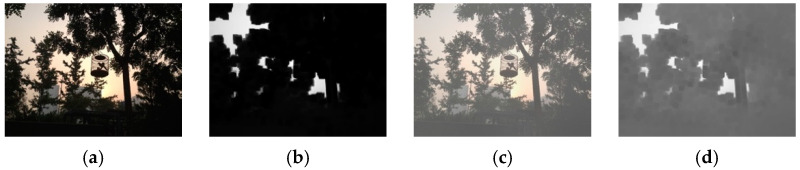
Example showing relation between haze and dark channel: (**a**) haze-free image; (**b**) dark channel of (**a**); (**c**) hazy image of (**a**); and (**d**) dark channel of (**c**).

**Figure 6 sensors-23-00815-f006:**
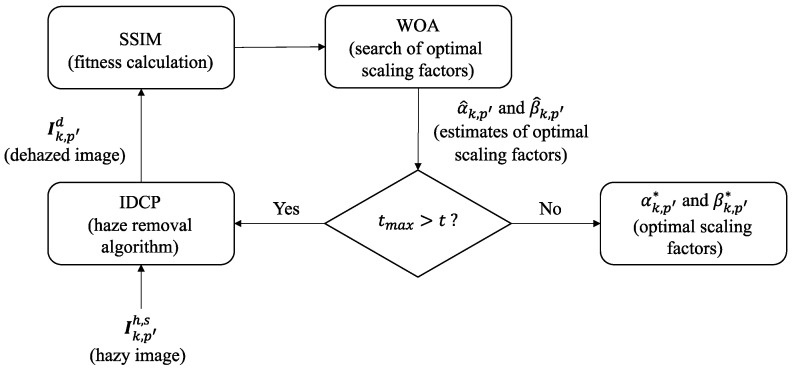
Block diagram of search for $\alpha_{k,i}^{*}$ and $\beta_{k,i}^{*}$ by IDCP/WOA.

**Figure 7 sensors-23-00815-f007:**
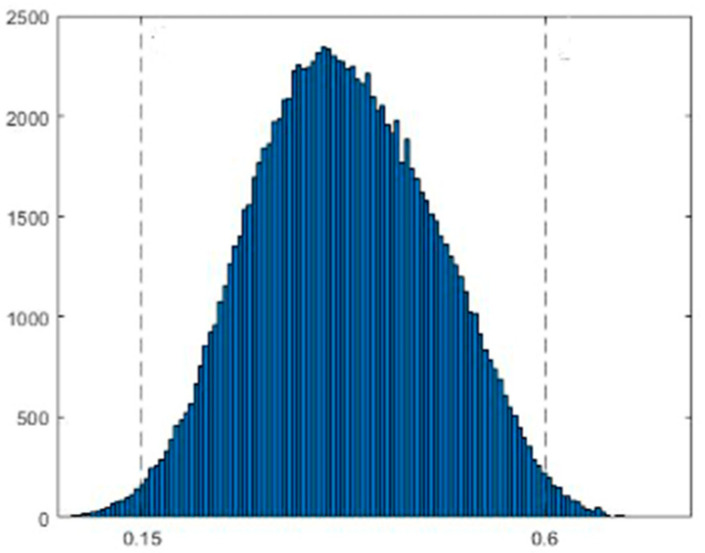
Distribution of ${\tilde{\mu }}_{dc}\left(0.9\right)$ in dataset S(0.05).

**Table 1 sensors-23-00815-t001:** Differences in parameters setting in DCP, IDCP, and IDCP/WOA.

Parameter		DCP	IDCP	IDCP/WOA
A		α=1 (fixed)	$\alpha_{a}$ (heuristic)	$\alpha^{*}$ (optimized by WOA)
$\tilde{t}\left(x\right)$		β=0.95 (fixed)	$\beta_{a}$ (heuristic)	$\beta^{*}$ (optimized by WOA)
GIF	guidance image	I 20 0.001	$I_{1}^{dark}(x)$ 55 0.1	$I_{1}^{dark}(x)$ 55 0.1
	N			
	ϵ			

**Table 2 sensors-23-00815-t002:** Five stages in proposed OIDCP and their functions.

Stage	Content	Function
1	Dataset S	Provides hazy–clear image pairs for WOA
2	HIC	Divides set S into subsets ${\hat{S}}_{k}$
3	HID	Screens hazy GT images in ${\hat{S}}_{k}$
4	IDCP/WOA	Searches $\alpha_{k,p^{\prime }}^{*}$ and $\beta_{k,p^{\prime }}^{*}$ for image pairs in $S_{k}$
5	OIDCP	Uses ${\overset{-}{\alpha }}_{k}^{*}$ and ${\overset{-}{\beta }}_{k}^{*}$ in application

**Table 3 sensors-23-00815-t003:** Dehazed images by IDCP/WOA with clear and hazy GT images.

GT Image	Hazy Image	IDCP/WOA	IDCP
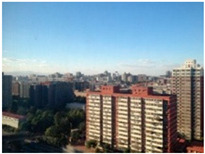	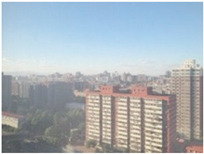	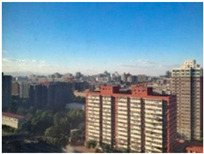	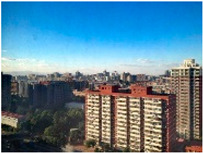
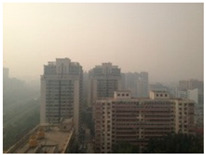	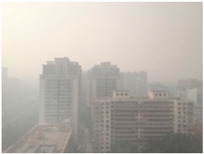	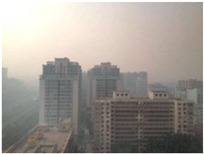	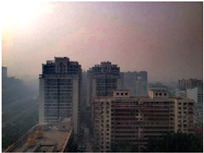

**Table 4 sensors-23-00815-t004:** Values of ${\tilde{\mu }}_{dc}\left(\tau \right)$ for clear and hazy GT images as τ varies from 0.9 to 0.4.

	$I_{1}$	$I_{2}$	$I_{3}$
Haze measure	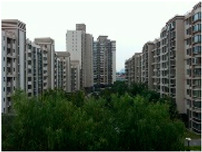	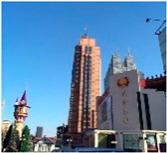	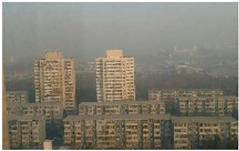
${\tilde{\mu }}_{dc}\left(0.9\right)$	0.0888	0.2742	0.3371
${\tilde{\mu }}_{dc}\left(0.8\right)$	0.0866	0.2742	0.3371
${\tilde{\mu }}_{dc}\left(0.7\right)$	0.0855	0.2742	0.3371
${\tilde{\mu }}_{dc}\left(0.6\right)$	0.0830	0.2721	0.3025
${\tilde{\mu }}_{dc}\left(0.5\right)$	0.0800	0.2613	0.2266
${\tilde{\mu }}_{dc}\left(0.4\right)$	0.0756	0.2461	0.1917
$∆{\tilde{\mu }}_{dc}$	0.0132	0.0281	0.1454

**Table 5 sensors-23-00815-t005:** Numbers of hazy and GT images in HID with various values of η.

η	0.025	0.05	0.075	0.1	Original
$N_{h}$	104,440	113,295	136,570	166,425	313,950
$N_{g}$	2984	3237	3902	4755	8970
R%	33.27	36.09	43.50	53.01	100

**Table 6 sensors-23-00815-t006:** HLs for various values of K and corresponding p%.

K						
HL k	5	6	7	8	9	10
1	(0,0.15]	(0,0.15]	(0,0.15]	(0,0.15]	(0,0.15]	(0,0.15]
	p%=0.7%	0.7%	0.7%	0.7%	0.7%	0.7%
2	(0.15,0.3]	(0.15,0.26]	(0.15,0.24]	(0.15,0.23]	(0.15,0.21]	(0.15,0.21]
	22.9%	12.4%	7.7%	5.3%	4.0%	3.1%
3	(0.3,0.45]	(0.26,0.37]	(0.24,0.33]	(0.23,0.3]	(0.21,0.28]	(0.21,0.26]
	52.0%	37.5%	25.5%	17.5%	12.5%	9.3%
4	(0.45,0.6]	(0.37,0.49]	(0.33,0.42]	(0.3,0.38]	(0.28,0.34]	(0.26,0.32]
	23.4%	34.5%	32.4%	27.0%	21.4%	16.8%
5	(0.6,1)	(0.49,0.6]	(0.42,0.51]	(0.38,0.45]	(0.34,0.41]	(0.32,0.38]
	1.1%	13.9%	23.4%	24.9%	23.5%	20.7%
6	-	(0.6,1)	(0.51,0.6]	(0.45,0.53]	(0.41,0.47]	(0.38,0.43]
	-	1.1%	9.2%	16.7%	19.3%	19.3%
7	-	-	(0.6,1)	(0.53,0.6]	(0.47,0.54]	(0.43,0.49]
	-	-	1.1%	6.7%	12.5%	15.2%
8	-	-	-	(0.6,1)	(0.54,0.6]	(0.49,0.54]
	-	-	-	1.1%	5.2%	9.7%
9	-	-	-	-	(0.6,1)	(0.54,0.6]
	-	-	-	-	1.1%	4.1%
10	-	-	-	-	-	(0.6,1)
	-	-	-	-	-	1.1%

**Table 7 sensors-23-00815-t007:** Comparison of PSNR and SSIM for IDCP with four metaheuristic optimization algorithms.

	IDCP/WOA	IDCP/MMFOA	IDCP/BRO	IDCP/MRFO
PSNR	28.526	28.270	27.997	28.209
SSIM	0.9403	0.9398	0.9384	0.9372

**Table 8 sensors-23-00815-t008:** Effect of $N_{s}$ and K on the IDCP/WOA (PSNR).

HL k											
K	$N_{s}$	1	2	3	4	5	6	7	8	9	10
5	2000 3000 4000 5000	30.78 30.78 30.80 30.84	30.39 30.39 30.39 30.47	28.74 28.93 28.95 28.91	26.25 26.21 26.35 26.36	23.32 23.61 23.31 23.99	-	-	-	-	-
6	2000 3000 4000 5000	30.71 30.66 30.62 30.79	30.50 30.61 30.48 30.57	29.64 29.68 29.75 29.71	27.97 27.93 27.95 27.88	25.76 25.78 25.93 25.84	23.88 24.01 23.82 23.70	-	-	-	-
7	2000 3000 4000 5000	30.95 30.84 30.82 30.89	30.64 30.60 30.50 30.55	30.26 30.23 29.98 29.96	28.95 28.92 28.77 28.96	27.15 27.09 27.21 27.14	25.54 25.46 25.35 25.48	23.63 23.72 23.76 23.55	-	-	-
8	2000 3000 4000 5000	30.82 30.82 30.82 30.71	30.66 30.65 30.65 30.67	30.16 30.26 30.26 30.31	29.65 29.49 29.49 29.48	28.14 28.08 28.08 28.13	26.59 26.50 26.50 26.64	25.25 25.16 25.26 25.24	23.74 23.77 23.64 24.00	-	-
9	2000 3000 4000 5000	30.86 30.81 30.63 30.69	30.82 30.73 30.71 30.61	30.39 30.44 30.27 30.38	29.91 29.81 29.87 29.81	29.01 29.05 28.79 28.91	27.59 27.59 27.51 27.58	26.43 26.45 26.30 26.39	25.25 25.16 25.26 25.24	23.74 23.77 23.64 24.00	-
10	2000 3000 4000 5000	30.56 30.57 30.53 30.82	30.55 30.52 30.78 30.70	30.58 30.52 30.78 30.70	29.89 29.92 30.13 30.12	29.59 29.48 29.50 29.60	28.37 28.35 28.29 28.45	27.30 27.28 27.44 27.31	26.17 26.17 26.16 26.07	25.29 25.30 24.93 24.85	23.87 23.84 23.47 23.84

**Table 9 sensors-23-00815-t009:** Objective evaluation of OIDCP and IDCP.

	OIDCP	IDCP
SSIM ↑	0.933 (1)	0.933 (1)
PSNR ↑	26.23 (1)	25.42 (2)
BRISQUE ↓	18.03 (2)	17.09 (1)
ILNIQE ↓	20.08 (2)	20.02 (1)
TMQI ↑	0.941 (1)	0.937 (2)
FSITM ↑	0.765 (1)	0.764 (2)
F&T ↑	0.853 (1)	0.851 (2)
$\overset{-}{R}$ ↓	1.286 (1)	1.571 (2)

**Table 10 sensors-23-00815-t010:** Subjective evaluation of IDCP and OIDCP.

	$I_{g}$	$I_{h}$	OIDCP	IDCP
$I_{1}$	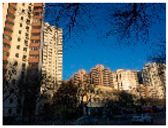	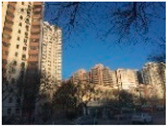 26.12	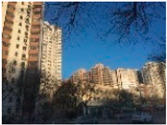 31.79	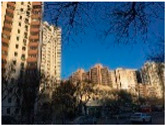 31.00
$I_{2}$	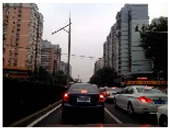	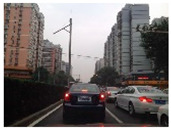 25.90	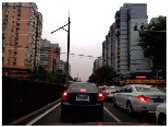 30.84	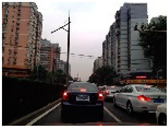 29.68
$I_{3}$	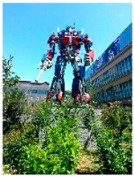	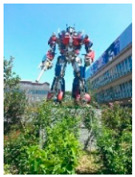 26.09	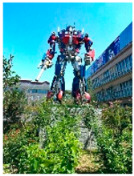 29.21	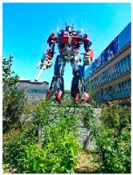 27.69
$I_{4}$	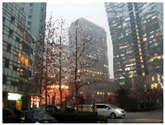	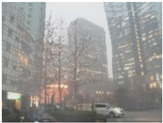 15.61	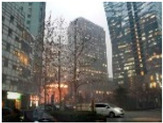 25.75	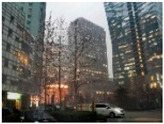 23.57
$I_{5}$	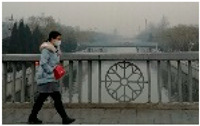	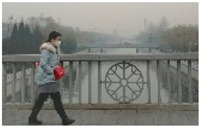 21.46	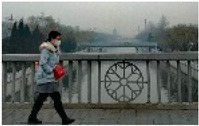 29.33	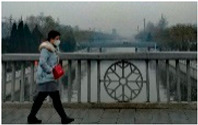 25.71
$I_{6}$	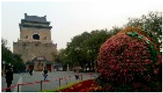	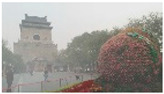 14.36	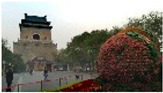 24.45	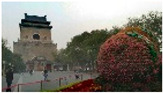 23.64
$I_{7}$	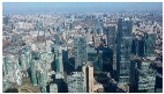	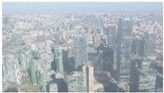 14.55	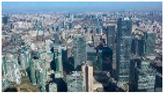 28.87	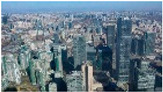 27.42
$I_{8}$	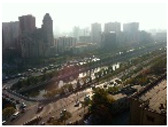	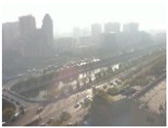 13.77	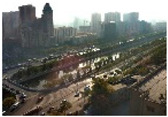 26.63	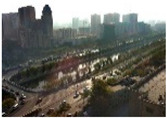 25.71
$I_{9}$	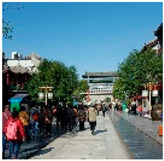	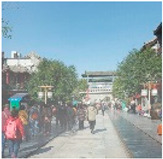 14.91	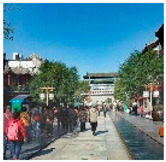 22.37	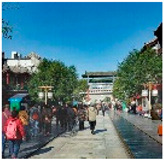 22.13
$I_{10}$	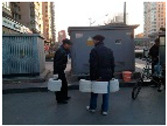	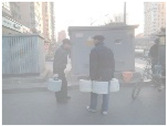 10.05	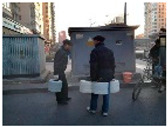 23.78	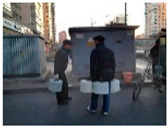 23.00

**Table 11 sensors-23-00815-t011:** Objective evaluation of comparison methods (RESIDE).

	OIDCP	IDCP	DCP	RRO	AOD	GCAN
SSIM ↑	0.933 (1)	0.933 (1)	0.878 (5)	0.890 (3)	0.886 (4)	0.911 (2)
PSNR ↑	26.23 (1)	25.42 (2)	18.20 (6)	20.95 (4)	20.63 (5)	24.96 (3)
BRISQUE ↓	18.03 (3)	17.09 (1)	17.73 (2)	18.64 (4)	20.82 (6)	20.49 (5)
ILNIQE ↓	20.08 (2)	20.02 (1)	20.55 (3)	20.81 (4)	23.87 (6)	21.24 (5)
TMQI ↑	0.941 (1)	0.937 (2)	0.869 (5)	0.917 (3)	0.906 (4)	0.917 (3)
FSITM ↑	0.765 (2)	0.764 (3)	0.763 (4)	0.772 (1)	0.735 (5)	0.772 (1)
F&T ↑	0.853 (1)	0.851 (2)	0.816 (5)	0.845 (3)	0.820 (4)	0.845 (3)
$\overset{-}{R}$ ↓	1.57 (1)	1.71 (2)	4.28 (5)	3.14 (3)	4.85 (4)	3.14 (3)

**Table 12 sensors-23-00815-t012:** Subjective comparison results (RESIDE).

	$I_{g}$	$I_{h}$	OIDCP	IDCP	DCP	RRO	AOD	GCAN
$I_{1}$	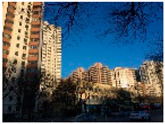	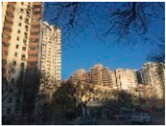	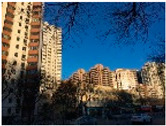	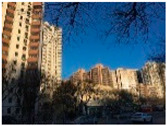	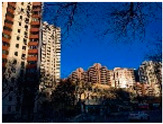	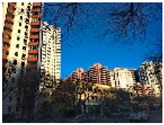	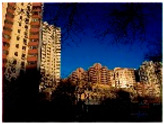	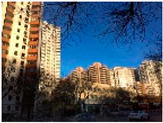
$I_{2}$	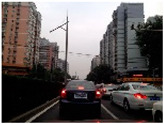	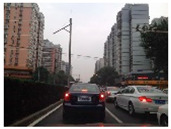	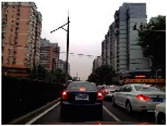	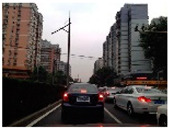	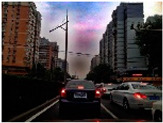	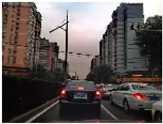	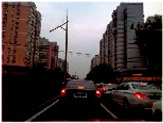	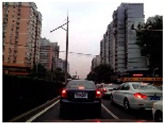
$I_{3}$	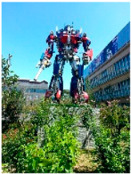	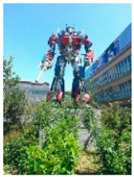	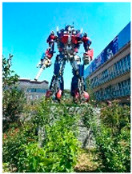	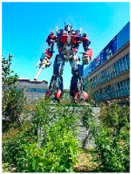	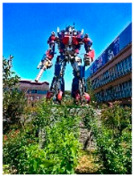	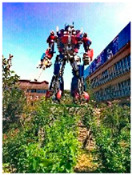	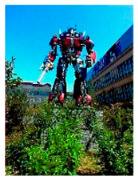	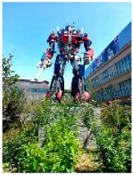
$I_{4}$	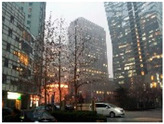	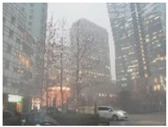	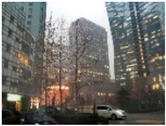	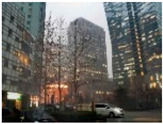	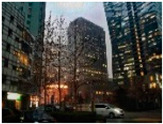	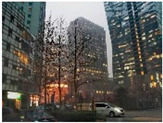	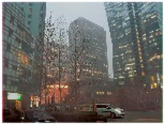	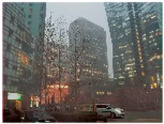
$I_{5}$	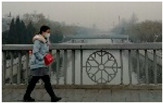	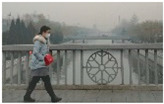	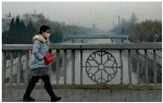	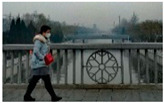	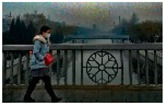	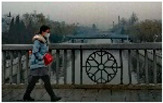	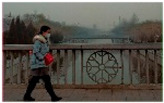	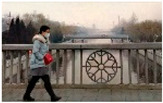
$I_{6}$	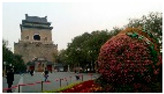	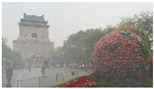	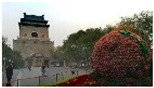	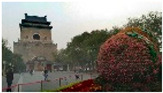	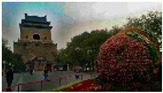	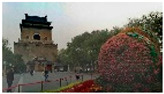	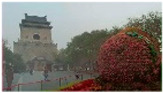	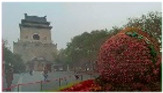
$I_{7}$	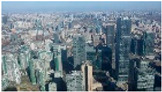	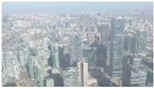	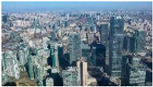	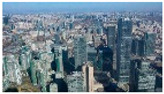	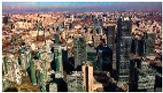	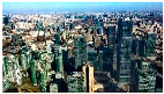	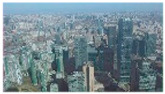	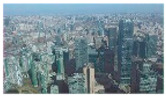
$I_{8}$	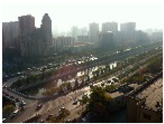	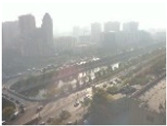	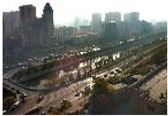	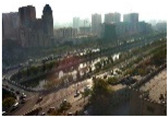	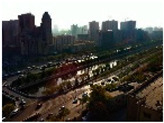	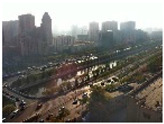	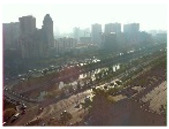	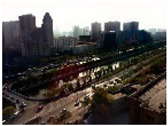
$I_{9}$	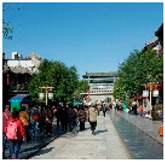	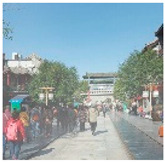	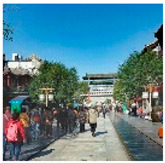	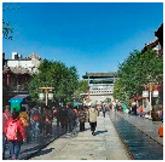	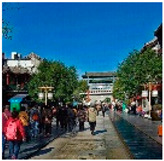	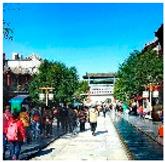	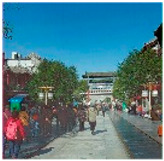	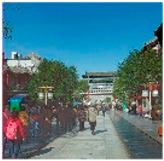
$I_{10}$	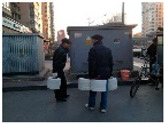	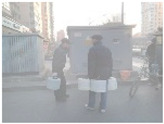	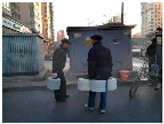	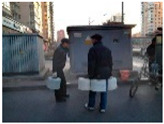	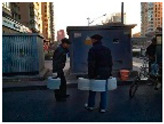	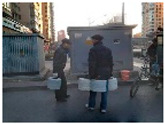	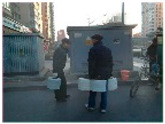	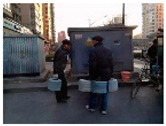

**Table 13 sensors-23-00815-t013:** Objective evaluation of comparison methods (O-HAZE).

	OIDCP	IDCP	DCP	RRO	AOD	GCAN
SSIM ↑	0.721 (1)	0.717 (2)	0.642 (6)	0.695 (4)	0.669 (5)	0.713 (3)
PSNR ↑	19.53 (1)	19.47 (2)	15.14 (6)	18.56 (4)	18.12 (5)	19.19 (3)
BRISQUE ↓	28.79 (5)	28.02 (4)	27.74 (3)	16.79 (1)	27.73 (2)	29.91 (6)
ILNIQE ↓	23.52 (3)	22.90 (2)	25.01 (4)	21.88 (1)	30.07 (5)	23.52 (3)
TMQI ↑	0.795 (2)	0.782 (3)	0.729 (5)	0.807 (1)	0.725 (6)	0.760 (4)
FSITM ↑	0.808 (1)	0.808 (1)	0.780 (4)	0.802 (2)	0.767 (5)	0.786 (3)
F&T ↑	0.801 (2)	0.795 (3)	0.754 (5)	0.804 (1)	0.746 (6)	0.773 (4)
$\overset{-}{R}$ ↓	2.14 (2)	2.43 (3)	4.71 (5)	2 (1)	4.85 (6)	3.71 (4)

**Table 14 sensors-23-00815-t014:** Subjective comparison results (O-HAZE).

Image	$I_{g}$	$I_{h}$	OIDCP	IDCP	DCP	RRO	AOD	GCAN
1	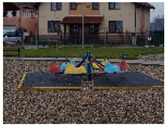	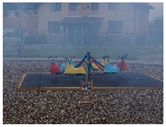	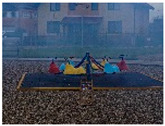	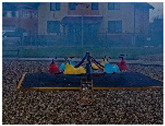	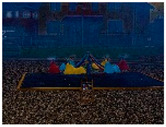	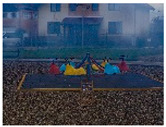	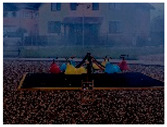	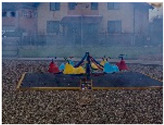
2	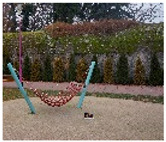	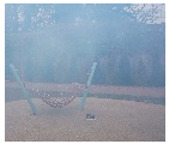	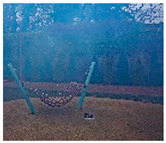	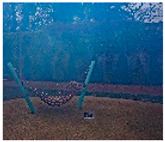	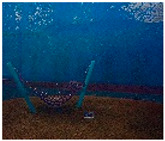	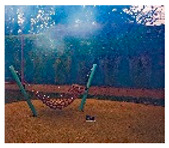	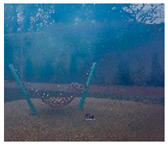	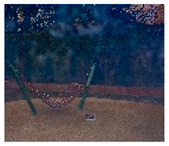
3	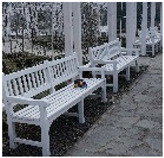	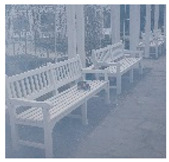	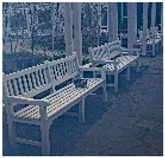	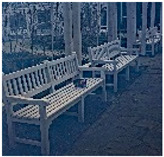	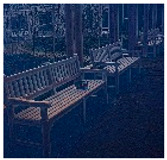	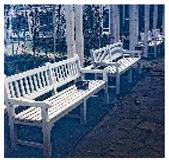	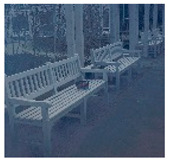	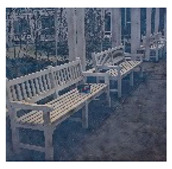
4	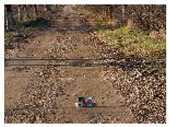	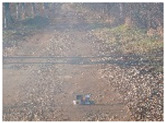	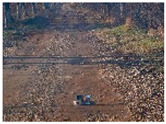	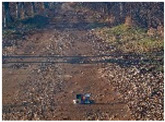	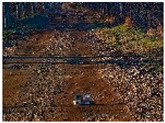	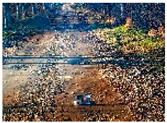	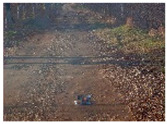	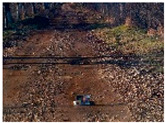
5	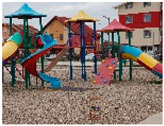	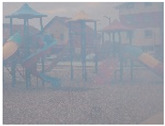	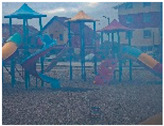	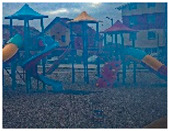	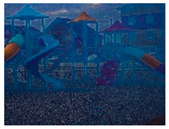	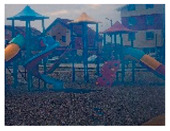	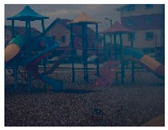	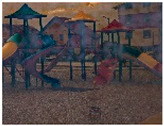

**Table 15 sensors-23-00815-t015:** Objective evaluation of comparison methods (KeDeMa).

	OIDCP	IDCP	DCP	RRO	AOD	GCAN
BRISQUE ↓	11.62 (2)	12.54 (5)	12.35 (4)	10.95 (1)	11.76 (3)	19.27 (6)
ILNIQE ↓	25.20 (3)	24.64 (2)	23.56 (1)	25.60 (4)	31.94 (6)	26.29 (5)
$\overset{-}{R}$ ↓	2.5 (1)	3.5 (3)	2.5 (1)	2.5 (2)	4.5 (4)	5.5 (5)

**Table 16 sensors-23-00815-t016:** Subjective comparison results (KeDeMa).

Image	$I_{h}$	OIDCP	IDCP	DCP	RRO	AOD	GCAN
1	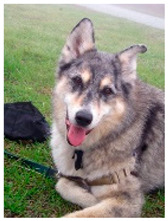	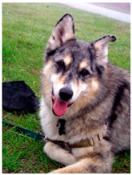	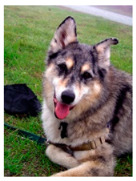	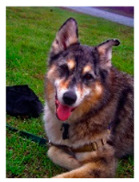	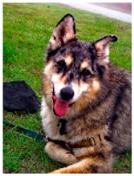	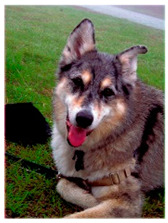	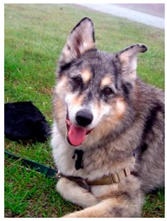
2	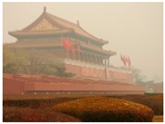	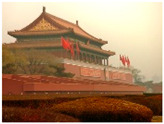	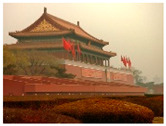	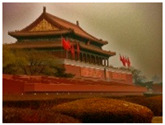	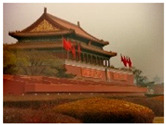	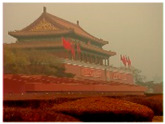	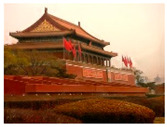
3	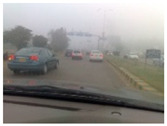	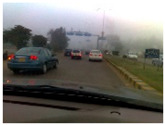	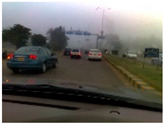	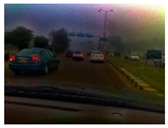	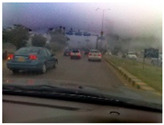	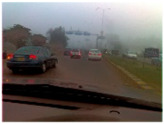	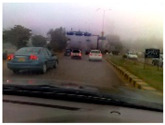
4	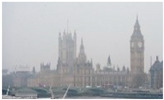	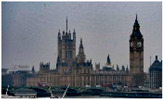	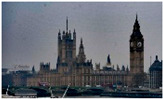	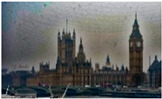	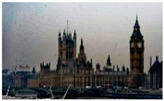	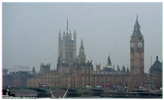	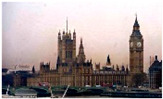
5	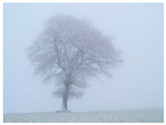	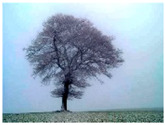	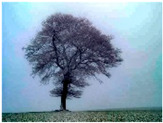	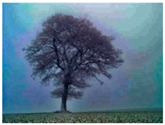	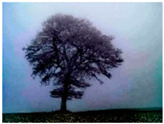	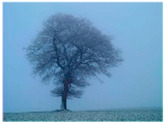	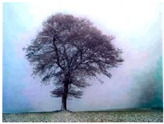

## Data Availability

The source codes for this study are available at https://github.com/chhsiehcyut/oidcp (accessed on 7 November 2022).

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
