# Peer review of "Using Whale Optimization Algorithm and Haze Level Information in a Model-Based Image Dehazing Algorithm"

_sensors, 2023, doi:10.3390/s23020815_

Round 1

Reviewer 1 Report

The presentation of this paper is simple and short. We recommend the author to improve the technical content and highlight the proposed methodology.

Related Works: This section required more attention and the author must include recent year papers related to different optimization algorithm and Dehazing Algorithm. Problem statements and limitations of the previous research must be discussed.

Dataset is limited, we strongly suggest you include a few more benchmark datasets. 

The author claimed that WOA and OIDCP gave an improvisation in terms of results. But the explanation is short and it's not reflected in the result section.

How does the author select this optimization, there is no baseline model comparison between those algorithms. How the best solution is derived (Voting, correlation or max). Explanation is required.

Result section discussed well but the importance of WOA and other optimization is missing. The results section should discuss the importance of individual algorithms and the other computational (hybrid) models.

Provide dataset and implementation code using third party repository / appendix for better understanding.

Reviewer 2 Report

1. In the abstract part, it is suggested that some data support should be added to ensure its integrity.

2. The introduction section of the paper refers to more detailed background information on the field, highlighting the work done by others, as well as their own innovations and improvements.

3. The block diagram in figure 3 is too simple. It is recommended to beautify the picture again.

4. The conclusion of this paper is lack of summary, and it is suggested to increase the research prospects of other cases.

5. It is suggested to quote the paper "Color image encryption algorithm based on DNA code and alternating quantum random walk".

6. A Complete Analysis of the Results and Study on the Limitations of Writing This Work.

Round 2

Reviewer 1 Report

Appreciate the efforts taken to revise the article and no more comments.